# Evolution of Resistance against Ciprofloxacin, Tobramycin, and Trimethoprim/Sulfamethoxazole in the Environmental Opportunistic Pathogen *Stenotrophomonas maltophilia*

**DOI:** 10.3390/antibiotics13040330

**Published:** 2024-04-05

**Authors:** Luz Edith Ochoa-Sánchez, José Luis Martínez, Teresa Gil-Gil

**Affiliations:** 1Centro Nacional de Biotecnología, Consejo Superior de Investigaciones Científicas (CSIC), Darwin 3, 28049 Madrid, Spain; luzedith.os@gmail.com; 2Department of Biology, Emory University, Atlanta, GA 30322, USA

**Keywords:** *Stenotrophomonas maltophilia*, antibiotic resistance, adaptative laboratory evolution

## Abstract

*Stenotrophomonas maltophilia* is an opportunistic pathogen that produces respiratory infections in immunosuppressed and cystic fibrosis patients. The therapeutic options to treat *S. maltophilia* infections are limited since it exhibits resistance to a wide variety of antibiotics such as β-lactams, aminoglycosides, tetracyclines, cephalosporins, macrolides, fluoroquinolones, or carbapenems. The antibiotic combination trimethoprim/sulfamethoxazole (SXT) is the treatment of choice to combat infections caused by *S. maltophilia*, while ceftazidime, ciprofloxacin, or tobramycin are used in most SXT-resistant infections. In the current study, experimental evolution and whole-genome sequencing (WGS) were used to examine the evolutionary trajectories of *S. maltophilia* towards resistance against tobramycin, ciprofloxacin, and SXT. The genetic changes underlying antibiotic resistance, as well as the evolutionary trajectories toward that resistance, were determined. Our results determine that genomic changes in the efflux pump regulatory genes *smeT* and *soxR* are essential to confer resistance to ciprofloxacin, and the mutation in the *rplA* gene is significant in the resistance to tobramycin. We identified mutations in *folP* and the efflux pump regulator *smeRV* as the basis of SXT resistance. Detailed and reliable knowledge of ciprofloxacin, tobramycin, and SXT resistance is essential for safe and effective use in clinical settings. Herein, we were able to prove once again the extraordinary ability that *S. maltophilia* has to acquire resistance and the importance of looking for alternatives to combat this resistance.

## 1. Introduction

*Stenotrophomonas maltophilia* is a cosmopolitan, ubiquitous, intrinsically multidrug-resistant Gram-negative bacterium with an environmental origin [1] that has been isolated in clinical [2,3,4,5,6] and non-clinical settings [7,8,9,10]. Its prevalence as a nosocomial pathogen increases every day, being mainly associated with respiratory infections in immunosuppressed and cystic fibrosis patients [7]. The increasing rate of antibiotic resistance has limited the therapeutic options and strategies to treat Gram-negative pathogens [11] such as *S. maltophilia*. This microorganism is considered a prototype of antibiotic-resistant bacteria. It exhibits an intrinsic low susceptibility to a wide variety of antibiotics such as β-lactams, aminoglycosides, tetracyclines, cephalosporins, macrolides, fluoroquinolones, and carbapenems [12]. For this reason, the World Health Organization (WHO) lists *S. maltophilia* as one of the main pathogens with interest in public health in hospitals worldwide [13]. Currently, the antibiotics’ combination trimethoprim/sulfamethoxazole (SXT) is the treatment of choice, with ceftazidime, ciprofloxacin, and tobramycin being the agents used in the majority of SXT-resistant infections [14,15,16]. With the number of antibiotics of choice restricted, predicting the mechanisms by which this bacterium may acquire resistance becomes important in preventing and treating infections [17,18]. *S. maltophilia* exhibits numerous mechanisms of antibiotic resistance that contribute to its multidrug-resistant phenotype, including the low permeability of its membrane as well as the presence in its genome of β-lactamases, enzymes that modify aminoglycosides, SmQnr (an enzyme that protects DNA gyrase from quinolones), and multidrug resistance (MDR) efflux pumps [19]. The primary cause of antibiotic resistance in this bacterial species is these efflux systems [20,21].

In the present work, we explore the evolution of resistance to ciprofloxacin (which inhibits bacterial topoisomerases, fundamental for DNA replication [22]), tobramycin (a ribosome-targeting antimicrobial), and SXT (an inhibitor of folic acid synthesis) in *S. maltophilia.* Ciprofloxacin is the most potent fluoroquinolone against Gram-negative bacilli [23]. The main mechanisms conferring resistance to this antibiotic are alterations in its target enzymes, the DNA topoisomerases GyrA and ParC [24], as well as the overproduction of MDR efflux pumps in most bacterial species [25]. Nevertheless, *S. maltophilia* is the only microorganism in which resistance to quinolones is not the consequence of mutations in the genes encoding these bacterial topoisomerases but of mutations that lead to the overexpression of efflux pumps [26]. Tobramycin is an aminoglycoside that induces miscoding during protein synthesis and the disruption of the bacterial membrane [27,28,29]. Resistance to tobramycin can result from different mechanisms involving mutations (like those leading to the overproduction of MDR efflux pumps), methylations, or enzymatic modifications of the antibiotic [30,31]. SXT is a fixed-dose combination antibiotic including sulfamethoxazole (a sulfonamide that inhibits folate synthesis) and trimethoprim (a direct competitor of the enzyme dihydrofolate reductase that produces a bactericidal effect) [32]. Bacterial resistance to SXT has been mainly attributed to the acquisition of resistance genes such as *dhfr*, *folP*, *sul1*, and *sul2*; the latter two are present in the core of ubiquitously distributed integrons. However, target mutations and mutations leading to the overproduction of MDR efflux pumps [33] also confer SXT resistance.

A previous analysis of single-step selected mutants in *S. maltophilia* detected that resistance to either ciprofloxacin or SXT was due to the overproduction of both the SmeDEF and the SmeVWX efflux pumps [33,34]. Nevertheless, this one-step selection only identifies mutations that can independently confer resistance to the antibiotic of selection. It does not provide information about the evolutionary dynamics, including the mutations (frequently low-level resistance mutations) that can jointly render resistance and those that can be relevant in clinics. For this purpose, experimental evolution and whole-genome sequencing (WGS) were used to examine the evolutionary trajectories of *S. maltophilia* toward resistance against these three antibiotics to determine the genetic changes underlying antibiotic resistance, as well as the evolutionary trajectories toward that resistance.

## 2. Results

### 2.1. Experimental Evolution in the Presence of Ciprofloxacin, Tobramycin, and Trimethoprim/Sulfamethoxazole Leads to High Levels of Resistance in S. maltophilia

To ascertain if distinct populations exhibit similar potential evolutionary trajectories, four biological replicates were subjected to selective pressure exerted by ciprofloxacin (CIP-A, CIP-B, CIP-C, and CIP-D), tobramycin (TOB-A, TOB-B, TOB-C, and TOB-D), and SXT (SXT-A, SXT-B32, SXT-B64, SXT-C, and SXT-D), and four were maintained without any selective pressure (A, B, C and D). All of them were serially passaged for 21 days, increasing the antibiotic concentration to track the progression of resistance during the selection period. The initial concentration of each antibiotic used in these experimental evolutions was the baseline minimal inhibitory concentration (MIC): 0.75 μg/mL of ciprofloxacin, 4 μg/mL of tobramycin, and 0.5 μg/mL of SXT. When bacteria are exposed to escalating antibiotic concentrations, one phenotypic trajectory can be anticipated: a gradual selection of mutants displaying progressively higher resistance levels. All the evolved populations, in the presence of the selective pressure exerted by each antibiotic, reached high levels of resistance (Table 1). All the populations that evolved in the absence of the drug had final MICs of 0.75 μg/mL to ciprofloxacin, 4 μg/mL to tobramycin, and 0.5 μg/mL to SXT.

### 2.2. Mutations Selected in the Presence of Ciprofloxacin, Tobramycin, and Trimethoprim/Sulfamethoxazole in S. maltophilia D457

To better understand the genetic events linked to the emergence of resistance in the evolved populations, the genome of each final population was sequenced. In addition to antibiotic resistance mutations, mutations increasing the fitness of the population for growing in the medium can also be selected. Therefore, only those mutations in the populations evolving under antibiotic selective pressure, but not in the control populations, were considered. Twenty-seven mutations were identified (Table 2 and Figure 1). Notably, three mutations were identified in different populations that evolved in the three antibiotics used. Firstly, insertions in the gene *rnE* that encodes the ribonuclease E. RnE plays a central role in RNA processing and metabolism [35], and it has been previously linked to high-level ciprofloxacin resistance in *Pseudomonas aeruginosa* and *Pseudomonas fluorescens* [36]. Secondly, three base pair insertions were found in the gene *SMD_4114.* This gene is proposed to encode an S9 family peptidase DAP2. Since endopeptidases have been predicted to function as space makers that trigger peptidoglycan enlargement due to the insertion of a new glycan strand and can be genetically associated with PBPs (penicillin-binding proteins), this mutation could be related to cross-resistance to beta-lactams (see below), such as ceftazidime [37]. Thirdly, single-nucleotide polymorphisms (SNP) or short deletions were detected in the gene *SMD_3479*, a YiiG family protein with unknown function and predicted to be a lipoprotein [38]. Another SNP was shared between ciprofloxacin and SXT-evolved populations in the gene *pip3*, a prolyl aminopeptidase. This protein is involved in the surveillance mechanism inducing the DNA-repair pathways [39].

Moving to ciprofloxacin-evolved populations, four mutations were exclusively identified in the populations that evolved in the presence of this drug. Genomic changes in the genes *soxR* (a redox-sensitive transcriptional activator that contributes to the multidrug resistant phenotypes of clinical strains) [40], *SMD_2503*, and *SMD_2704* (both of unknown function), and *smeT* were found. SmeT is a regulator of *smeDEF* expression. Mutations in this regulator deal to the overproduction of the SmeDEF efflux pump, and hence to MDR in this bacterial species [19].

Twelve genetic changes were found exclusively in the tobramycin-evolved populations. SNPs in the gene that encodes the L1 50S ribosomal protein, *rplA*, were detected, which is consistent with the tobramycin mechanism of action. Again, mutations related to the SmeDEF efflux pump were found; in this case, an SNP in the gene encoding SmeD, the periplasmic adaptor subunit of the multidrug efflux transporter SmeDEF. Furthermore, we identified mutations in genes not previously related to aminoglycosides resistance: *glpG*, which encodes an intramembrane serine protease of the rhomboid family; *rsmB*, an RNA regulator that interacts with *rsmA*, and which overexpression increases the production of N-acyl-homoserine lactone, pyocyanin, and elastase [41]; *motB*, which encodes a protein that integrates into the cell membrane and is part of the flagellar motor protein complex [42]; *parE*, encoding the DNA topoisomerase IV subunit B and whose mutation renders to quinolone resistance in *E. coli* [43], *S. typhi* [44] or *R. anatipestifer* [45]; *spoT*, a synthetase-hydrolase that regulates the concentration of (p)ppGpp [46]; *SMD_3194*, an ATP-binding protein of unknown function; *SMD_3405*, a putative membrane protein of unknown function; *SMD_1169*, an acetyltransferase; *SMD_2317*, an ABC transporter; and *SMD_2955*, an PepSY-associated TM helix domain-containing protein that has been suggested to function as a controller of peptidase activity within the immediate environmental and, in addition, protect the cell from lysis.

Finally, seven mutations were found in the SXT-evolved populations. Importantly, this includes an SNP in the gene *smeRv*, a transcriptional regulator whose mutation leads to the overproduction of the SmeVWX efflux pump, whose contribution to the acquisition of resistance to SXT in single-step selected mutans was previously described in *S. maltophilia* [47]. Additionally, an SNP in the gene *folP*, that encodes a dihydropteroate synthase, the target enzyme of the sulfonamides, was detected. This enzyme confers sulfonamide resistance by preventing the inhibition of folate synthesis by sulfonamide antibiotics, such as SXT [48]. Another SNP in the gene *SMD_3621*, a pteridine reductase related to the synthesis of folates in bacteria, was also found [49]. Moreover, mutational changes in *pstS* (the substrate-binding component of the ABC-type transporter complex *pstSACB*, involved in phosphate import [50]), *cblD* (a pilus assembly protein that is required for surface expression of cable pili but is not related to antibiotic resistance [51]), and *SMD_1644* and *SMD_2325* (two hypothetical proteins with unknown function and for which participation in antibiotic resistance has not been reported yet) were identified.

Among the mutations, there were 20 SNPs, 4 insertions, and 3 deletions. Notably, almost all mutant alleles selected in the presence of antibiotics had coverages of >90%, (Table 2). The identified mutations show the versatility of the resistance mechanisms of *S. maltophilia*. We identified mutations in genes encoding RND efflux pumps, oxidative stress response proteins, outer membrane regulators, resistance regulators, and virulence determinants. Below, we discuss their functions according to our results.

### 2.3. Adaptative Trajectories, Cross-Resistance, and Collateral Sensitivity of Evolved Populations

To assess whether the development of antibiotic resistance was specific to the antibiotic used for selection or impacted the susceptibility to other antibiotics, the resistance levels to other antibiotics were measured. Nine families of antibiotics (beta-lactams, fluoroquinolones, tetracyclines, macrolides, aminoglycosides, polymyxins, phenols, monobactam, and phosphonic) were tested. Almost all evolved populations demonstrated increased resistance or susceptibility to other antibiotics from various structural families, indicating that some resistance mutations are not specific to ciprofloxacin, tobramycin, and SXT (Figure 2).

The evolved populations in ciprofloxacin (CIP-A, CIP-B, CIP-C, and CIP-D) showed cross-resistance to tetracycline, nalidixic acid, and ofloxacin and collateral susceptibility to SXT and tobramycin. The four tobramycin-evolved populations (TOB-A, TOB-B, TOB-C, and TOB-D) displayed cross-resistance to tetracycline and collateral sensibility to chloramphenicol, erythromycin, and ciprofloxacin. Additionally, two tobramycin-evolved populations (TOB-B, TOB-D) were hypersusceptible to SXT.

Concerning evolved populations in SXT, three populations (SXT-A, SXT-C, and SXT-D) were unable to acquire high levels of resistance to SXT but presented cross-resistance to tigecycline, tetracycline, aztreonam, nalidixic acid, or colistin. Furthermore, the two populations acquiring high levels of SXT resistance (SXT-B32 and SXT-B64) showed cross-resistance to ciprofloxacin, ofloxacin, aztreonam, nalidixic acid, tetracycline, and chloramphenicol. These results are consistent with the mutation in *smeRV* identified in these two populations that would lead to an overproduction of the SmeVWX efflux pump, which contributes to the acquisition of resistance to the aforementioned antibiotics [33]. SXT-B32 and SXT-B64 also demonstrated collateral susceptibility to tobramycin, streptomycin, ceftazidime, fosfomycin, polymyxin B, and colistin.

## 3. Discussion

In this work, we identified that all the evolved populations reached high levels of resistance in the presence of the selective pressure exerted by each antibiotic, in comparison with the populations that evolved in the absence of the drugs. Since the purpose of this work was to identify stable mutations that can be fixed, only the final evolved populations were sequenced. We are aware that sequencing intermediate evolved populations will also provide information on the dynamics of the evolution of antibiotic resistance; however, this study is beyond the focus of the current work. Twenty-seven mutations were identified. Among them, twenty correspond to SNPs, three to deletions, and four to insertions. The mutations affect elements of the outer membrane, oxidative stress response, previously known resistance determinants, and virulence. We also studied the cross-resistance and collateral sensitivity of these evolved populations.

Among the mutations found during the course of these evolutions, insertions in the gene *rnE* (Ribonuclease E) were identified in populations selected in the presence of the three antibiotics. This gene plays a central role in RNA processing and metabolism [35], and it is emerging as a potential antibacterial target in *Acinetobacter baumanni* [52]. *rnE* is required for the maturation of the 5S and 16S rRNAs and the majority of tRNAs, including the mRNA processing and cleaving of the 5’ leader of the *ompA* mRNA [53]. We have found that the *S. maltophilia* genome encodes a porin orthologous to the *Escherichia coli* OpmA and the *P. aeruginosa* OprF (SMD_2502) [54]. Clinical isolates of *P. aeruginosa* that are antibiotic-resistant to imipenem and polymyxin B and deficient in the major outer membrane protein OprF have been isolated in a previous work [36]. This result suggests that the lack of processing of the OpmA mRNA would lead to a decrease of this porin in the outer membrane, increasing antibiotic resistance to multiple antibiotics. This might suggest that OmpA/OprF could be a major outer membrane antibiotic transporter in *S. maltophilia* as it is in *P. aeruginosa* [36], since its blockage is enough to confer high-level ciprofloxacin resistance. This resistance could lead to erythromycin susceptibility too, leading to increased binding of this antibiotic to its target site in the 50S ribosomal subunit, since RnE is required for the maturation of rRNAs.

Ciprofloxacin-evolved population CIP-D presented an SNP in a gene encoding a protein that shows 95% identity with YadA, an adhesin precursor from *P. aeruginosa*. This collagen-binding outer membrane protein forms a fibrillar matrix on the bacterial cell surface that promotes initial attachment and invasion of eukaryotic cells. Although it also protects the bacteria by being responsible for agglutination, serum resistance, complement inactivation, and phagocytosis resistance, this change has not been related to antibiotic resistance.

Populations CIP-B and CIP-C shared an SNP in *soxR*. This redox-sensitive transcriptional activator induces the expression of the RND efflux pump-encoding operon *mexGHI-opmD* in *P. aeruginosa* [55]. Moreover, the constitutive *soxS* expression caused by single point mutations in the *soxR* gene has been shown to contribute to the MDR phenotypes of clinical strains, and it is sufficient to confer multiple-antibiotic resistance in a fresh genetic background. The increased *soxS* expression in *E. coli* leads to the downregulation of the expression of the gene encoding the outer membrane porin OmpF [56,57], to a decrease in cell permeability, and to an increased expression of the genes encoding the AcrAB efflux pump [58]. All in all, *soxRS*-mediated antibiotic resistance is a result of the combination of an increased efflux pump activity and decreased cell permeability [40]. Hence, our results indicate that SoxR is important for ciprofloxacin resistance in *S. maltophilia*, as has been previously described for other organisms like *A. baumannii* [59], *E. coli* [40], and *K. pneumoniae* [60].

A SNP in the *smeDEF* regulator, *smeT*, was found in a ciprofloxacin-evolved population (CIP-B). Since SmeDEF is a main determinant of MDR in *S. maltophilia*, the observed cross-resistance to tigecycline, chloramphenicol, erythromycin, SXT, or tetracycline is consistent with the fact that these antibiotics are substrates of SmeDEF [61]. Further, the same substitution, L166Q, has been previously associated with antibiotic resistance in clinical isolates of *S. maltophilia*, supporting the reliability of our results [61].

All in all, these results suggest that decreased permeability and the overproduction of MDR efflux pumps are the main mechanisms driving ciprofloxacin resistance, as well as the broad cross-resistance caused by this selection.

Regarding tobramycin-evolved populations, populations TOB-A, TOB-B, and TOB-C presented different SNPs in the gene *rplA*, encoding the 50S ribosomal protein L1, which has been previously related to the aminoglycosides’ resistance in *S. maltophilia* [62].

The populations TOB-B and TOB-D showed molecular changes in *spoT*. SpoT regulates the nutritional starvation stringent response [63,64,65]. Although this synthetase-hydrolase of the alarmone ppGpp has not been previously related to antibiotic resistance, it has been proposed that the stringent response can modulate antibiotic resistance and tolerance [66]. It is important to notice that the population TOB-D only presents the *spoT* mutation and that the coverage of the mutation was 100%. The fact that no further mutations are found in the evolved population strongly suggests that this mutation is responsible for the acquired tobramycin resistance in this population, although more work would be needed to fully support this statement.

In the population TOB-A, an SNP was found in a gene related to metabolism: *glpG.* This gene encodes a rhomboid family intramembrane serine protease required to produce an extracellular signaling molecule that regulates cellular functions, including peptidoglycan acetylation, methionine transport, and cysteine biosynthesis. Previously, a *glpG* mutant of *E. coli* exhibited a slight increase in resistance to β-lactams [67]. Interestingly, the tobramycin-evolved population TOB-A presents greater cross-resistance levels to ceftazidime than the other populations that evolved on tobramycin.

Another important change in this TOB-A population is an insertion in the gene encoding the ribosomal RNA small subunit methyltransferase B (RsmB), a regulatory RNA of the global repressor RsmA in *P. aeruginosa* [41]. RsmA regulates the Type III Secretion System (T3SS) in *P. aeruginosa*. This system has been proposed to play a role in the expression of MDR efflux pumps in *P. aeruginosa*. A reduction in T3SS expression in this bacterial species is associated with the overproduction of MexCD-OprJ and MexEF-OprN [68]. Hence, compared to the *P. aeruginosa* wild-type strain, the *rsmA* mutant presents increased resistance to amikacin, nalidixic acid, trimethoprim, gentamicin, and ceftazidime. The confirmed role of RsmA in antibiotic resistance implies that RsmA could be a possible global regulator involved in regulating the cross-talk between antibiotic resistance and the virulence associated with T3SS [69]. Thus, this mutation could lead to an unregulated RsmA that produces increased antibiotic resistance.

Furthermore, we identified an SNP in *smeD* in the population TOB-B. To date, only mutations in the SmeDEF regulator protein (SmeT) have been described to be related to an MDR phenotype. However, it is also known that mutations involving the subunits of efflux pumps in this bacterial species (changes in the SmeH structural element of the SmeGH efflux pump) are involved in the acquisition of resistance [70], suggesting that this mutation could be related to an enhanced efflux of antibiotics such as ofloxacin or tetracycline.

Moving to SXT-evolved populations, we achieved a final concentration of 3MIC in three populations (A, C, and D) and final concentrations of 32MIC (SXT-B32) and 64MIC (SXT-B64) in two other populations. These technical issues might indicate that the acquired resistance to this antibiotic is complex and dependent on a leading mutation. The two highly resistant populations shared a mutation in the *smeRv* regulator. This mutation would lead to an overproduction of the SmeVWX efflux pump, which contributes to the acquisition of resistance to SXT, ciprofloxacin, ofloxacin, nalidixic acid, levofloxacin, tetracycline, and chloramphenicol [33], antibiotics to which these two populations are cross-resistant.

Population SXT-C presented a mutation in the gene *pstS*, the substrate-binding component of the ABC-type transporter complex PstSACB involved in phosphate import [50]. The Pst system encoded by the *pst* operon (*pstSCAB-phoU*) forms a phosphate transporter across the cytoplasmic membrane. Mutations in this operon have been shown to influence antibiotic susceptibility to polymyxin in *E. coli*. This effect is due to changes in the expression of RND, MFS, and ABC transporters influenced by *pstC* disruption [71]. In addition, mutations in this gene significantly decrease bacterial adherence, invasion, motility, and biofilm-forming ability in *A. baumannii* [72].

Population SXT-D presented a mutation in the gene *folP* that encodes a dihydropteroate synthase, the target enzyme of sulfonamide. As mentioned, this enzyme prevents the inhibition of folate biosynthesis by sulfonamide antibiotics, such as the one included in SXT, thus conferring sulfonamide resistance. Previous studies related *folP* point mutations with SXT resistance in *Streptococcus mutans* [48].

In order to raise high-level SXT resistance, the population SXT-B64 presented a mutation in two hypothetical proteins with unknown function, SMD_1644 and SMD_2325, and in a pteridine reductase. It has been previously studied that overproduction of the pteridine reductase 1 (Ptr1) by gene amplification confers methotrexate resistance in *Leishmania promastigotes* [73]. However, the reasons why mutations in this gene are selected by SXT in *S. maltophilia* remain to be clarified.

Finally, in some of the populations involved in ciprofloxacin (CIP-C and CIP-D) and SXT (SXT-A, SXT-B32, and SXT-B64) resistance, mutations in the gene *pip3*, encoding a prolyl aminopeptidase, were detected. Pip3 is a regulator of a major facilitator antiporter involved in pristinamycin resistance in *Streptomyces coelicolor* [39]. Whether or not it plays a similar role in *S. maltophilia* antibiotic resistance, regulating the expression of a drug efflux pump, remains to be established.

In conclusion, resistance to all these antibiotics is related to permeability changes and overexpression of the genes encoding MDR efflux pumps. This result would indicate the need to introduce new antibiotics or new combinatory therapies, together with the development of efflux pump inhibitors, to treat *S. maltophilia* [19] since high-level resistance is rapidly acquired. Significantly, all the populations that evolved in ciprofloxacin showed collateral susceptibility to tobramycin, and all populations that evolved in tobramycin were susceptible to ciprofloxacin. Reciprocal collateral sensitivity is not a frequent situation [74]. When found, it favors the use of combinations of the antibiotics involved. Our findings support that the sequential combinatory use of ciprofloxacin and tobramycin might improve the treatment outcome of *S. maltophilia* infections. Nevertheless, we are aware that the translation of these results into clinical practice requires the analysis of the robustness of the observed evolution pathways in clinical isolates presenting different genomic backgrounds [74], a study that is beyond the purposes of the current work.

## 4. Materials and Methods

### 4.1. Bacterial Strains and Growth Conditions

The wild-type clinical isolate *S. maltophilia* D457 was used as the parental strain for the evolution experiments [75]. All the experiments were performed at 37 °C in Mueller–Hinton broth with shaking at 250 rpm in glass tubes.

### 4.2. Experimental Evolution

Experimental evolution was performed with the wild-type strain D457 [75] growing in the presence of increasing concentrations of ciprofloxacin, tobramycin, and SXT. Sixteen independent bacterial populations (four controls without antibiotics, four populations challenged with ciprofloxacin, four populations challenged with tobramycin, and four populations challenged with SXT). Cultures were grown in parallel in Mueller-Hilton broth at 37 °C and 250 rpm in independent glass tubes. Cultures were initially grown at the maximum concentration of the antibiotics that allowed growth in Mueller–Hinton broth. Serial passages were performed by inoculating 1 μL of bacterial cell cultures in fresh medium containing the same antibiotic concentration every 24 h for 2 days. The initial concentrations used were 0.5 μg/mL ciprofloxacin, 0.5 μg/mL tobramycin, and 0.25 or 0.5 μg/mL SXT. Every three days, the concentration of the drugs was doubled. Ciprofloxacin and tobramycin concentrations increased over the evolution from the initial MIC up to 32MIC. As stated, MIC is defined as the lowest concentration of an antibiotic that inhibits the growth of a specific bacterial strain [76]. From SXT, 5 populations were started in 0.5 μg/mL SXT (3 grew until a final concentration of 3MIC and 1 until 32MIC). An extra population started at 0.25 μg/mL was grown until a final concentration of 64MIC. Every three days, samples from each culture were taken and preserved at −80 ◦C for future investigation. The procedure was repeated for 21 consecutive days.

### 4.3. DNA Extraction and Whole-Genome Sequencing

At the end of the evolution assays, the total genomic DNAs from the evolved populations were extracted. The genomic DNA extraction was performed with the Chemagic DNA H96 bacterial kit (CMG-799 Chemagic) using the Chemagic 360/MSMI instrument (PerkinElmer, Waltham, MA, USA). The DNA quality assay was performed with an Agilent 2200 TapeStation system from the Translational Genomics Unit (Ramón y Cajal Institute in Madrid, Spain). Library construction and WGS were performed by the Oxford Genomics Centre (Oxford, United Kingdom). Pair End Libraries (2 × 150 bp) were sequenced using an Illumina NovaSeq6000 system (San Diego, CA, USA). The coverage was greater than 150× for all the samples.

### 4.4. Identification of Mutations

The variant calling and VCF (Variant Call Format) file were created with Bactmap v1.0. [77] against the reference genome of the strain *S. maltophilia* D457 (GenBank accession number NC_017671.1). BCFtools v1.19. [78] were used for filtering and binding individual CVFs. Variants were then filtered against the D457 laboratory wild-type strain.

### 4.5. Antimicrobial Susceptibility Assays

MICs for the antibiotics ciprofloxacin, tobramycin, SXT, tigecycline, streptomycin, tetracycline, ofloxacin, aztreonam, nalidixic acid, ceftazidime, chloramphenicol, fosfomycin, and erythromycin were determined on Mueller–Hinton agar plates. MIC of colistin and polymyxin B were determined on Mueller–Hinton II agar plates. MIC test strips (Liofilchem, Roseto degli Abruzzi, Italia) were used in all the cases, and the plates were incubated for 20 h at 37 °C.

## Figures and Tables

**Figure 1 antibiotics-13-00330-f001:**
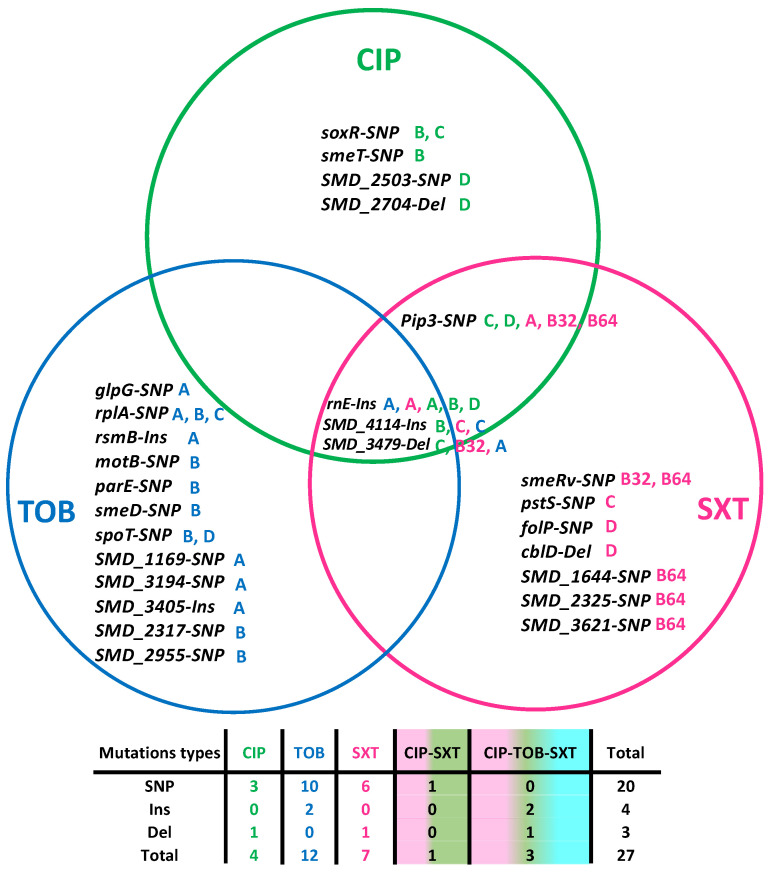
Distributions of mutations in the evolved lineages: twenty-seven mutations were identified. Among them, three were found in populations that evolved in the three antibiotics used. One extra mutation was shared between ciprofloxacin (CIP) and SXT and the other were specific for the antibiotic used for selection. Four were found in ciprofloxacin-evolved populations, twelve in the tobramycin (TOB) populations, and seven in the SXT. The white boxes indicate the population in which that mutation was found. The table below indicates the type of mutation according to the antibiotic in which the populations evolved. SNP: single-nucleotide variant, Ins: insertions, Del: deletions. Green A, B, C, and D correspond to the CIP-evolved populations in the antibiotics CIP; Blue A, B, C, and D correspond to the TOB-evolved population; and A, B32, B64, C, and D to the SXT-evolved populations.

**Figure 2 antibiotics-13-00330-f002:**
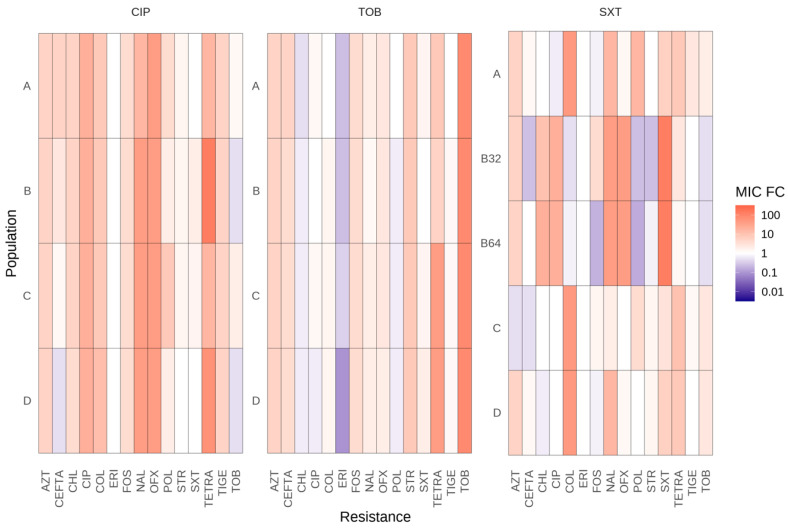
Collateral susceptibility of the evolved *S. maltophilia* populations to antibiotics from different families. MICs fold changes in the 17 fosfomycin-evolved populations, with respect to the populations that evolved in the absence of antibiotics. Values of at least double or half the control populations’ MIC were considered significant. CIP, ciprofloxacin; TOB, tobramycin; SXT, trimethoprim/sulfamethoxazole; TIGE, tigecycline; STR, streptomycin; TETRA, tetracycline; OFX, ofloxacin; AZT, aztreonam; NAL, nalidixic acid; CEFTA, ceftazidime; CHL, chloramphenicol; FOS, fosfomycin; ERI, erythromycin; COL, colistin; POL, polymyxin B. A, B, C, and D correspond to the CIP and TOB-evolved populations in the first two panels. A, B32, B64, C, and D to the SXT-evolved populations in the third panel.

**Table 1 antibiotics-13-00330-t001:** MICs (μg/mL) of the seventeen populations evolved in the presence of the selective pressure exerted by each antibiotic. The populations reached high levels of resistance in comparison with the populations that evolved in the absence of the drugs.

	Population	Day 0	Day 3	Day 6	Day 9	Day 12	Day 15	Day 18	Day 21
Ciprofloxacin	CIP-A	0.75	6	8	>32	>32	>32	>32	>32
	CIP-B	0.75	8	8	>32	>32	>32	>32	>32
	CIP-C	0.75	8	8	>32	>32	>32	>32	>32
	CIP-D	0.75	8	8	>32	>32	>32	>32	>32
Tobramycin	TOB-A	4	16	>256	>256	>256	>256	>256	>256
	TOB-B	4	16	>256	>256	>256	>256	>256	>256
	TOB-C	4	>256	>256	>256	>256	>256	>256	>256
	TOB-D	4	48	>256	>256	>256	>256	>256	>256
Sulfamethoxazole-trimethoprim (SXT)	SXT-A	0.5	0.5	0.75	1.5	1.5	1.5	1.5	1.5
	SXT-C	0.5	0.5	0.75	1.5	1.5	1.5	1.5	1.5
	SXT-D	0.5	0.5	0.75	1.5	1.5	1.5	1.5	1.5
	SXT-B32	0.5	0.5	0.75	1.5	>32	>32	>32	>32
	SXT-B64	0.25	0.5	0.75	1.5	>32	>32	>32	>32

**Table 2 antibiotics-13-00330-t002:** WGS-identified mutations in the ciprofloxacin, tobramycin, and SXT-evolved lineages.

	L	Gene	Product	Localization	Type	Nucleotide Change	Amino Acid Change	Frequency (%)	Domain
Ciprofloxacin	A	*rnE*	Rne/Rng family ribonuclease	3139248	Ins	A⟶ACCGAGCTGGGTG	N486Fs	98	Ribonuclease E
	B	*soxR*	Redox-sensitive transcriptional activator	1129982	SNP	C⟶T	R45W	100	Helix-Turn-Helix DNA binding
		*smeT*	Efflux transporter SmeDEF transcriptional repressor	4099641	SNP	T⟶A	L166Q	100	PRK10668 DNA binding
		*rnE*	Rne/Rng family ribonuclease	3139248	Ins	A⟶ACCGAGCTGGGTG	N486Fs	99	Ribonuclease E
		*SMD_4114*	S9 family peptidase	4619837	Ins	A⟶AGTG	H773Fs	99	DAP2 peptidase
	C	*pip3*	Prolyl aminopeptidase	830195	SNP	C⟶A	A46N	57	
		*soxR*	Redox-sensitive transcriptional activator	1129982	SNP	C⟶T	R45W	90	Helix-Turn-Helix DNA binding
		*SMD_3479*	YiiG family protein	3881832	Del	GGGA⟶G	P195Fs	99	DUF3829
	D	*pip3*	Prolyl aminopeptidase	830195	SNP	C⟶A	A46N	47	
		*SMD_2503*	ESPR-type extended signal peptide-containing protein	2791884	SNP	A⟶C	G893G	96	
		*SMD_2704*	Hypothetical protein	3006994	Del	CAAACA⟶C	Q276Fs	99	
		*rnE*	Rne/Rng family ribonuclease	3139248	Ins	A⟶ACCGAGCTGGGTG	N486Fs	80	Ribonuclease E
Tobramycin	A	*glpG*	Rhomboid family intramembrane serine protease	412958	SNP	T⟶C	T17A	100	Membrane-associated serine protease
		*rplA*	50S ribosomal protein L1	884707	SNP	T⟶G	F22C	100	Ribosomal L1 bact
		*SMD_1169*	GNAT family N-acetyltransferase	1297246	SNP	A⟶G	Q28P	95	C0G3818 acetyltransferase
		*SMD_3194*	ATP-binding protein	3550846	SNP	C⟶G	R1017P	9	
		*rnE*	Rne/Rng family ribonuclease	3139248	Ins	A⟶ACCGAGCTGGGTG	N486Fs	90	Ribonuclease E
		*SMD_3405*	DUF2339 domain-containing protein	3798031	Ins	C⟶CTCTGGCGGCCGG	A47Fs	99	DUF2339
		*SMD_3479*	YiiG family protein	3881834	Del	GAAT⟶G	T194Fs	100	DUF3829
		*rsmB*	16S rRNA (cytosine(967)-C(5))-methyltransferase	4243528	Ins	G⟶GC	R191Fs	90	PRK10901
	B	*rplA*	50S ribosomal protein L1	885157	SNP	A⟶G	H172R	93	Ribosomal L1 bact
		*motB*	Flagellar motor protein	1081733	SNP	A⟶T	L197Q	95	
		*parE*	DNA topoisomerase IV subunit B	1823237	SNP	G⟶T	R291Q	94	PRK05559
		*SMD_2317*	ABC transporter six-transmembrane domain-containing protein	2575602	SNP	G⟶A	R287Q	92	
		*SMD_2955*	PepSY-associated TM helix domain-containing protein	3271344	SNP	A⟶C	A466A	91	
		*smeD*	Multidrug efflux RND transporter periplasmic adaptor subunit	4098218	SNP	G⟶A	Q235K	92	PRK15030
		*spoT*	Bifunctional (p)ppGpp synthetase/guanosine-3’,5’-bis(diphosphate) 3’-pyrophosphohydrolase	3846537	Ins	A⟶ACAGGCGGCG	T712Fs	99	SpoT superfamily
	C	*rplA*	50S ribosomal protein L1	884715	SNP	G⟶A	A25T	73	Ribosomal L1 bact
		*SMD_4114*	S9 family peptidase	4619837	Ins	A⟶AGTC	H773Fs	99	DAP2 peptidase
	D	*spoT*	Bifunctional (p)ppGpp synthetase/guanosine-3’,5’-bis(diphosphate) 3’-pyrophosphohydrolase	3847032	SNP	C⟶T	G547S	100	SpoT superfamily
SXT	A	*pip3*	Prolyl aminopeptidase	830195	SNP	C⟶A	A46N	70	
		*rnE*	Rne/Rng family ribonuclease	3139248	Ins	A⟶ACCGAGCTGGGTG	N486Fs	99	Ribonuclease E
	B32	*pip3*	Prolyl aminopeptidase	830195	SNP	C⟶A	A46N	55	
		*smeRv*	LysR family transcriptional regulator	1936539	SNP	C⟶T	G266D	100	C-terminal domain of LysR
		*SMD_3479*	YiiG family protein	3881831	SNP	G⟶T	P195H	100	DUF3829
	B64	*pip3*	Prolyl aminopeptidase	830195	SNP	C⟶A	A46N	53	
		*SMD_1644*	DUF47 family protein	1818431	SNP	T⟶G	L205W	100	YkaA
		*smeRv*	LysR family transcriptional regulator	1936735	SNP	T⟶A	N201K	100	C-terminal domain of LysR
		*SMD_2325*	Hypothetical protein	2584531	SNP	T⟶C	N50E	100	
		*SMD_3621*	Pteridine reductase	4056328	SNP	T⟶G	V179G	96	PRK09135
	C	*pstS*	Phosphate ABC transporter substrate-binding protein	1551867	SNP	C⟶T	W215R	92	Periplasmic Binding Protein Type 2
		*SMD_4114*	S9 family peptidase	4619837	Ins	A⟶AGTG	H773Fs	99	DAP2 peptidase
	D	*folP*	Dihydropteroate synthase	1841316	SNP	G⟶C	G151A	87	DHPS
		*cblD*	CfaE/CblD family pilus tip adhesin	3834921	Del	ATGTACTT⟶A	Q206Fs	99	

L: lineage; SNP: single-nucleotide polymorphism; Ins: insertion; Del: deletion; fs: frame shift; Frequency (%): percentage of reads that contain the variation within a heterogeneous population; ND: non-determined (IGV Genomics Software v2.9.0 does not determine the frequency of insertion/deletions).

## Data Availability

All the data generated are available in this manuscript. Copies of the genome sequence here have been made available under the NCBI BioProject ID PRJNA933546.

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
