# Peer review of "Evolution of Resistance against Ciprofloxacin, Tobramycin, and Trimethoprim/Sulfamethoxazole in the Environmental Opportunistic Pathogen *Stenotrophomonas maltophilia"

_antibiotics, 2024, doi:10.3390/antibiotics13040330_

Round 1

Reviewer 1 Report

Comments and Suggestions for Authors

Dear Editor,
The manuscript "Evolution of resistance against ciprofloxacin, tobramycin, and trimethoprim/sulfamethoxazole in the environmental opportunistic pathogen Stenotrophomonas maltophilia" by Luz Edith Ochoa-Sánchez and co-authors described Stenotrophomonas maltophilia's antibiotic resistance and evolution. The results show how the strain acquires resistance, which is intriguing to the readers. However, a few questions need be addressed before consideration for publishing.

1.The manuscript only utilized one strain of Stenotrophomonas maltophilia D457; what about additional strains of S. maltophilia? Do you have comparable resistance characteristics? Perhaps more strains of S. maltophilia should be used in the paper.
2. Please define the MIC in the method section.
3.Please adjust the phrase in L15-L16.
4.Please double-check the concentrations of all antibiotic agents; the unit used in the article differs from your MIC values, such as in L144-L145.
5.There were a few inaccuracies in the manuscript, including L66, L147, L151, and so on.

Author Response

1.The manuscript only utilized one strain of Stenotrophomonas maltophilia D457; what about additional strains of S. maltophilia? Do you have comparable resistance characteristics? Perhaps more strains of S. maltophilia should be used in the paper. 

We appreciate the comment of the referee. When looking to the bibliography, usually this type of studies are performed using a model strain to detect novel mutations. In few occasions, this is followed by studies focusing on the robustness of the evolution pathways, a topic that is beyond the purposes of the current work, using of a set of isolates  (15-20) presenting different genomic backgrounds. The referee would be aware that this will require a 15-20 fold increase in the resources and time required for the analysis; more than a simple modification required for confirming the results  presented in the article, this is an ambitious new project focusing on robustness that, as stated above is not the topic of the current work. Nevertheless, since robustness is important for the application of the results (particularly collateral sensitivity), a new paragraph discussing this issue has been added.

  1. Please define the MIC in the method section.

Done

3.Please adjust the phrase in L15-L16. 

Done.

4.Please double-check the concentrations of all antibiotic agents; the unit used in the article differs from your MIC values, such as in L144-L145. 

We use µg/mL all along the article.

5.There were a few inaccuracies in the manuscript, including L66, L147, L151, and so on.

The article has been revised to detect inaccuracies

Reviewer 2 Report

Comments and Suggestions for Authors

In the following manuscript, the authors aim to evaluate antibiotic resistance mechanisms for resistance against ciprofloxacin, tobramycin, and 2 trimethoprim/sulfamethoxazole in the environmental opportunistic pathogen Stenotrophomonas maltophilia. The authors relied on whole genome sequencing of strains that survived following antibiotic exposure to evaluate these mechanisms. I would like to suggest the following revisions:

1. Include gram negative bacteria in the introduction section.

2.  Explain why MIC readings over 32 for cipro and SXT and readings over 256 for tobramycin weren’t reported.

3. Expand term PBP in line 117.

4. Rellying on whole genome sequencing data is not sufficient to state that mutation in SpoT is responsible for tobramycin resistance in TOB-B and TOB-D populations. Since SpoT is not previously linked to antibiotic resistance, the only way to conclusively prove that mutations in this gene lead to antibiotic resistance, the authors must generate spoT knockout derivatives of a wild-type strain and show that following gene deletion, antibiotic resistance is observed.

Comments on the Quality of English Language

Editing is required. Certain sentences are structured incorrectly and hence are difficult to understand. 

Author Response

  1. Include gram negative bacteria in the introduction section. 

Done

  1. Explain why MIC readings over 32 for cipro and SXT and readings over 256 for tobramycin weren’t reported. 

The concentration ranges correspond to clinically relevant concentrations. We used E-test strips to follow the evolution test. The reason why the E-test strips for CIP and SXT have a range of 0.002 to 32 μg/ml, while those for tobramycin have a range of 0.016 to 256 μg/ml, is related to the intrinsic sensitivity of the microorganisms to these antibiotics and the concentrations necessary to inhibit their growth. Tobramycin usually requires higher concentrations to be effective against certain microorganisms than ciprofloxacin. Therefore, tobramycin E-test strips have a wider concentration range to cover the concentrations needed to inhibit the growth of different bacterial strains.

  1. Expand term PBP in line 117. 

Done

  1. Rellying on whole genome sequencing data is not sufficient to state that mutation in SpoT is responsible for tobramycin resistance in TOB-B and TOB-D populations. Since SpoT is not previously linked to antibiotic resistance, the only way to conclusively prove that mutations in this gene lead to antibiotic resistance, the authors must generate spoT knockout derivatives of a wild-type strain and show that following gene deletion, antibiotic resistance is observed.

We appreciate the comment of the referee because this forced us to search the bibliography in more detail. When doing that, we found that while SpoT mutations have not been associated to antibiotic resistance, it has been described that alterations in the stringent response, which is regulated by SpoT, modulate antibiotic tolerance and resistance. This information is now discussed in the article. Regarding the proposed experiment, the referee would like taking into consideration that spoT has a dual role, synthetizing and degrading the alarmone ppGpp. The mutation in the population TOB_D is a G547S change; we cannot know if this produces the inactivation or a change in the activity of SpoT. Instead of deleting spoT as suggested by the referee, we would need to re-create the same mutation (with the risk of selecting compensatory mutations in the process) and sequence the constructed mutant; the same situation that the referee is concerned with regarding the population. Since the population contains just this mutation and the mutation is in 100 % of the population, and the article does not focus on spoT, we believe that the best option is to discuss that the presence of this unique mutation in population TOB-D "strongly suggest" that is associated to resistance, but that more work is still needed to confirm this statement.

Reviewer 3 Report

Comments and Suggestions for Authors

This study explores the evolutionary trajectories towards antibiotic resistance in Stenotrophomonas maltophilia, specifically against ciprofloxacin, tobramycin, and trimethoprim/sulfamethoxazole (SXT), using experimental evolution and whole-genome sequencing (WGS). It finds that mutations in efflux pump regulatory genes smeT and soxR are crucial for ciprofloxacin resistance, mutations in rplA for tobramycin resistance, and mutations in folP and efflux pump regulator smeRV for SXT resistance. This research enhances our understanding of the genetic mechanisms underpinning antibiotic resistance in S. maltophilia, highlighting the organism's remarkable ability to acquire resistance and underscoring the need for alternative treatment strategies.

Clarifications in the manuscript:

Table 1 - The value of 0.75 µg/ml mentioned for populations without antibiotics presumably represents the baseline MIC. But the manuscript didn’t mention what it is. Clarification on this baseline would help interpret the data more accurately.

Line 191- "MICS" should be corrected to "MICs".

The study provides new information on the genetic mutations leading to antibiotic resistance in S. maltophilia, while a few questions need to be addressed,

1.        To capture a more dynamic picture of resistance evolution, it might be beneficial to conduct WGS at various stages of antibiotic exposure, not just the final step. This could unveil a more detailed timeline of mutation emergence and fixation.

2.        Single colony selection: The study used the samples from culture for WGS, while that generated mixed populations. It might gain more accuracy analysis from a isolating single colonies on agar plates with varying antibiotic concentrations. This approach could uncover clonal resistance patterns that might be obscured in a heterogeneous population.

3.        The complexity of antibiotic resistance mechanisms suggests a multifaceted interplay between different mutations and compensatory adaptations that may influence resistance. Future studies should consider this complexity to fully understand the resistance phenotype.

Author Response

Table 1 - The value of 0.75 µg/ml mentioned for populations without antibiotics presumably represents the baseline MIC. But the manuscript didn’t mention what it is. Clarification on this baseline would help interpret the data more accurately.

Thanks for the observation. Correction is made in the text.

Line 191- "MICS" should be corrected to "MICs".

Done

The study provides new information on the genetic mutations leading to antibiotic resistance in S. maltophilia, while a few questions need to be addressed,

  1. To capture a more dynamic picture of resistance evolution, it might be beneficial to conduct WGS at various stages of antibiotic exposure, not just the final step. This could unveil a more detailed timeline of mutation emergence and fixation.

Thank you for your valuable comment. We agree that a detailed analysis of emergence and evolution along time of the mutations is of interest for understanding the dynamics of evolution. However, being interesting, this study is a full new project by itself that will require a large amount of resources and time and is beyond the purposes of the work.  What we wanted to determine are the stable mutations that can accumulate after antibiotic challenge. That's why we focused on sequencing the last step. This feature is now discussed in the article.

  1. Single colony selection: The study used the samples from culture for WGS, while that generated mixed populations. It might gain more accuracy analysis from a isolating single colonies on agar plates with varying antibiotic concentrations. This approach could uncover clonal resistance patterns that might be obscured in a heterogeneous population.

We appreciate the comment of the referee because it states a critical aspect in these studies. It is better to perform experimental evolution experiments and sequence all the population or is it better plating directly on selective plates and sequencing the selected mutants? The answer is that each approach serves for a different purpose. Experimental evolution as performed here, under increasing antibiotic concentrations, serve to select mutations that in combination give increased antibiotic resistance and, eventually, compensatory mutations. Direct plating in selective plates (as we have published in different articles), select single mutations that can confer resistance without the need of another mutation. As stated in the introduction, we have already performed the experiment suggested by the referee and have determined the mechanisms of resistance that are selected after direct plating and colony selection in the case of ciprofloxacin and SXT, being the only mechanisms selected under this experimental setup overexpression of the efflux pumps SmeDEF and SmeVWX (refs 26, 33). The study here performed hence has served to increase the knowledge in the mechanisms of resistance to quinolones of S. maltophilia.

  1. The complexity of antibiotic resistance mechanisms suggests a multifaceted interplay between different mutations and compensatory adaptations that may influence resistance. Future studies should consider this complexity to fully understand the resistance phenotype.

Yes. We consider future studies using strains with different genetic backgrounds, sequencing of different steps and studies on the impact of resistance in bacterial physiology.  However these studies are beyond the purposes of the current work.

Round 2

Reviewer 1 Report

Comments and Suggestions for Authors

Dear Editor,

The authors submitted a revised manuscript taking into account all the concerns. I think the authors have made a great effort and the manuscript can be accepted by Antibiotics.

Reviewer 2 Report

Comments and Suggestions for Authors

I am satisfied with the answers to the queries I raised after reviewing the original manuscript. This version can now be accepted for publication. 

Comments on the Quality of English Language

Minor editing is required

Reviewer 3 Report

Comments and Suggestions for Authors

Thank you for the comprehensive revisions and the detailed response to the comments. I appreciate the thoroughness with which you have revised and providing clearer insight into your research process. The changes made have addressed the previous concerns effectively.